# Bullying prevalence in Pakistan’s educational institutes: Preclusion to the framework for a teacher-led antibullying intervention

**Sohni Siddiqui*, Anja Schultze-Krumbholz**

Department of Educational Psychology, Technische Universität Berlin, Berlin, Germany

* s.zahid@campus.tu-berlin.de












**Data Availability Statement:** All relevant data are within the paper.

## Abstract

Increasing reports of bullying and cyberbullying in schools in recent years are undeniable and have been recognized as a serious public health problem. Conventional bullying and cyberbullying are not only a problem in higher educational institutions in Pakistan, but also in primary and secondary schools. Although statistics show higher levels of bullying and cyber-risky behaviors among youth, policies and interventions to control the consequences of conventional and cyberbullying are rare in the Pakistani context. This study explores teachers’ perspectives and experiences in identifying bullying strategies in different school contexts. Four hundred fifty-four teachers working in different educational institutions completed an online survey that provided data to draw conclusions and to get a better sense of the situation in educational institutions in Pakistan. According to the results, teachers experience verbal and social bullying more frequently than online and physical bullying. In addition, teachers in lower grades reported noticing more physical bullying than teachers in higher grades. Facebook was reported to be the most common platform students used to bully each other. Researchers also found significant differences between rural and urban teachers’ experiences with social bullying. Bullying intervention strategies should be developed and integrated into educational settings in Pakistan. The data presented will be used to develop tailored anti-bullying interventions that are culturally and socially appropriate for Pakistani educational settings.

## Introduction

Aggression is a set of actions that are considered a significant challenge for society to deal with and are defined by social psychology as any behavior aimed at harming a person or animal [1]. If there is no immediate intervention, some of these aggressive behaviours, can lead to serious societal consequences, such as extreme forms of bullying or irreversible negative effects such as delinquency [2]. Bullying is a repeated aggressive behavior with the intention to afflict physical, emotional, or mental harm and usually results from a power imbalance [3]. To differentiate, bullying others without using electronic or digital means is nowadays considered to be “traditional bullying”; when technology is used to intentionally harm others, it is referred to as

**Funding:** The funders had no role in study design, data collection and analysis, decision to publish, or preparation of the manuscript.

**Competing interests:** The authors have declared that no competing interests exist.

cyberbullying. School bullying and neighborhood bullying are two examples of traditional bullying, while cyberbullying is a result of technology, such as the Internet [4].

Traditional bullying, or face-to-face bullying, is further subdivided into physical, verbal, or social/emotional/relational bullying. Acts such as hitting, kicking, tripping, pinching, and pushing or damaging property are considered different forms of physical bullying. Contrarily, name-calling, insults, use of swear words, teasing, intimidation, and rude remarks are different kinds of verbal abuse or verbal bullying. Social bullying, sometimes referred to as covert/relational or emotional bullying, is often challenging to recognize and can be carried out behind the bullied person's back. It is designed to harm someone's social reputation and/or to cause them humiliation. The most common types of this behavior include lying and spreading rumors, negative facial or physical gestures, menacing or contemptuous looks, playing nasty jokes to embarrass and humiliate, mimicking unkindly, encouraging others to socially exclude someone, damaging someone's social reputation, or affecting acceptance of the person [5].

The advent of technology and frequent internet usage has introduced a more technologically-oriented form of aggression known as cyber aggression [6]. It is defined as 'intentional harm delivered by the use of electronic means to a person or a group of people irrespective of their age who perceive(s) such acts as offensive, derogatory, harmful or unwanted' [7]. Englander et al. (2017) defined cyberbullying as willful and repeated harm inflicted upon a victim through the use of computers, cell phones, and other electronic devices [8]. Even though several initiatives and interventions have been designed to prevent and control this harmful behavior, the rate of cyberbullying and traditional bullying continues to rise around the world [9]. Facebook statistics revealed that between 2017 and 2021, harassment-related posts constantly increased [10]. Delgado (2020) also reported that abusive language among children and teens rose by 70% and more soon after engagement in online classes [11]. Moreover, as school work had to be done from home, traditional bullying was replaced by cyberbullying in households [12].

The issue of bullying and cyberbullying has gained prominence worldwide, but for educational institutions in Pakistan, this matter has yet to be studied in depth. Additionally, the vast majority of the data collected and published are based on respondents' self-reports, which may be skewed by social desirability biases or misremembrance issues, thus undermining validity. The aim of this article is to discuss the range of traditional bullying and cyberbullying and their concomitants in Pakistan's educational system by including experiences of teachers working at different educational institutions. Moreover, the need for a socially and culturally adapted/newly developed bullying intervention will be discussed by revisiting steps and interventions taken in the past and the data gathered from teachers.

## Theoretical background

### The theoretical basis for explaining the concept of bullying and aggression

Aggression is a broad term comprising multiple constructs and is not just limited to behavior evaluated at the symptom level [13]. Aggression is also described from a variety of perspectives including antisocial behavior, juvenile delinquency, coercion, assertiveness, or bullying [14]. Gay (1999) summarized the concept of the Psychoanalytical Theory presented by Sigmund Freud in the late 19th century and stated that aggression is an innate and fundamental feeling which is a part of human nature; it is essential for defense and the fight for dominance [15]. Pinker (2012) supported Freud's concept and explained that the motive behind bullying is the need for power, which is a special kind of instrumental violence that is inherent in human nature [16, 17]. Dehue (2013); Slonje *et al.*, (2013) reported that traditional bullying and cyberbullying behaviors constitute unjustified aggression, based on power imbalance, and continue

over time [18, 19]. A psychoanalytic model explains that human nature is always seeking superiority, and bullying is a way of displaying authority and showcasing prestigiousness.

In early studies, one of the other main factors being considered was frustration as a cause of aggression [20]. Similarly, Agnew (1992) developed a General Strain Theory and defined strain as events or conditions that individuals dislike, with strains also classified as the inability to achieve desired goals, the presentation of harmful or negatively valued stimuli, and the loss of desirable stimuli [21]. All these strains can be seen as major contributors to a perpetrator's involvement in bullying behavior and victims' transformation into perpetrators. The literature has elaborated that bullying and cyberbullying are generally associated with people who have been maltreated or have a higher level of anxiety, academic difficulties, passive aggressive behaviors, and internalizing and externalizing problems than their peers [22–24].

Albert Bandura introduced the concept of social learning in the development of aggression. He further elaborated that interventions designed to modify behavior via rewards and punishment were incomplete for explaining the development of behaviors, as humans tend to mimic others and learn from them [25]. Garandeau and Cillessen (2006) explained that perpetrators are popular and considered to have competent social-cognitive skills [26]. The desire for supremacy and high status are some of the motives behind bullying others [27]. Bystanders often mimic bullies to gain the same social influence as they perceive a bully's fame and peers' anxiety about becoming the next victim.

Another theory regarding child behavior is the Social Information Processing Theory (SIP), which examines how children and teenagers process information in social contexts. According to SIP, children with disruptive behavior problems perceive, interpret, and make decisions about social information in a way that increases their likelihood of engaging in aggressive behavior [28]. Attachment problems or coercive cycles could explain such difficulties with interpersonal processing. Similarly, coercive parenting can explain harsh parental behavior such as hitting, yelling, scolding, threatening, rejection, and psychological control to achieve compliance from the child. As an example of how social information processing theory is applied, a child may assume another child intentionally pushed them in the lunch line rather than assuming it was an accident.

Social Interaction Theory proposed by Tedeschi and Felson (1994) demonstrated aggression as psychologically influenced behavior aimed at changing the target's behavior [29]. These actions are used by a perpetrator to obtain something valuable such as money, goods, information, services, or to achieve desired social or self-identity. Anderson and Bushman (2002) introduced the General aggression model (GAM) that integrated existing mini-theories of aggression into an amalgamated whole. This model is more comprehensive, explains aggressive acts based on multiple motives, and provides broader insights into child-rearing and development issues [30].

## Status and frequency of bullying and cyberbullying in Pakistan

The increase of bullying and cyberbullying in academic settings in recent years is indisputable [31] and has been established as a serious public health problem [32], with long-term negative effects on physical and mental health [33]. Similarly, traditional and cyberbullying have not only found their way into Pakistan's higher educational institutions, but appear in primary and secondary schools as well [34, 35]. Various investigations have shown that bullying and cyberbullying are common practices in Pakistan's educational institutions and have affected the physical, emotional, and mental health of students.

Saleem et al. (2021) report that the level of cyberbullying has substantially increased in educational institutions in Pakistan [35]. Data gathered from universities of the province of Sindh

has confirmed that cyberbullying is common in urban universities. Previously, Musharraf and Anis-ul-Haque (2018) also supported the findings of [35] and found that more than 60% of university students were involved in cyberbullying behavior [36]. Similarly, Mirza *et al.* (2020) found that cyberbullying is ubiquitous in higher educational institutions [37].

Saleem et al. (2021) added that substantial differences in victimization and perpetrators were found with respect to socioeconomic status and access to the Internet [35]. Further, Rafi (2019) reports that linguistic skills were exploited by the aggressors to victimize the participants [38]. Young social media addicts often have offline disputes, which becomes the rudimental rationale for cyber-associated behaviour [38–40]. Although studies concluded that boys are more involved in perpetration and victimization, researchers have reported that females have also been victimized through conventional and online media. Magsi et al. (2017) found that females in universities are also being scoffed at and harassed using electronic media, but about half of the victims do not disclose this due to cultural and religious restraints and to protect themselves from being blamed as immoral [41]. Women suffer in silence and as a self-defense leave activities that are taking play in cyberspace. Lack of knowledge about how to handle cyberbullying and lack of trust in law enforcement agencies are additional important factors that encourage bullies to victimize women in urban university settings. It is also reported that females are more susceptible to developing anxiousness due to cyber victimization as compared to their male counterparts [36]. Additionally, both targets of bullying and offenders of bullying experienced adverse emotional and social consequences. Bullying perpetrators exhibited a greater severity of depressive symptoms due to problems in psychosocial functioning [42].

Bullying and cyberbullying is not limited to the university level but have permeated the schooling system in Pakistan [34, 42–45]. Khawaja et al. (2015) found that violence in the form of physical and verbal abuse is commonplace in major cities and provincial capitals [44]. Asif (2016) further added that bullying and victimization are also associated with poor academic performance [46] and they are one of the causes of the high dropout rate in schools [45].

Murshid (2017) and Musharraf and Anis-ul-Haque (2018) reported that victimization is the major cause of mental health issues, such as anxiety and depression among youth in Pakistan, while low to middle-income countries like Pakistan have limited resources to address such mental issues [47, 48]. It is recommended in recent publications [6, 35, 49, 50] to build support centers in academic settings to deal with bullying and cyberbullying situations and to implement anti-bullying interventions. The goal of these centers is to raise students' awareness of prevention and coping measures. In Pakistan, interventions should be tailored to the country's specific circumstances.

## Sources of frustration-aggression in Pakistani society

Dollard et al. (1939) considered that frustration and dissatisfaction are the main causes of aggression development [51]. In continuation, the concept of displaced aggression described by Denson et al. (2006) also explains how the level of frustration redirects aggression to an alternative target to cope with stress [52]. Moreover, Patchin and Hinduja (2011) explained bullying behaviour in terms of the General Strain Theory (Agnew, 1992) that argues that individuals who experience strain feel angry or frustrated as a result and are more at risk to engage in criminal, deviant or bullying behavior [21, 53]. Correspondingly, traditional bullying and cyberbullying are more common among people who are traditionally or cyber victimized, show a high level of anxiety, academic difficulties, passive-aggressive behaviours, and internalizing and externalizing problems than among their peers [22–24]. Husain (2000) reported that post-independence economic development has predominantly benefited a small class of the

elite, while the majority of the population remains uneducated and poor [54]. Unemployment, accelerating inflation, uncontrolled population growth and low literacy rates are some additional enduring factors in the declining standard of living of Pakistan's major population [55, 56]. Rapid urbanization, limited and insecure water supply [57], food insecurity and malnutrition [58] are some of the additional factors that contribute to rising aggression in Pakistan's society. Empirically, it is reported that there is a high prevalence of behavioral problems and emotional and behavioral difficulties among Pakistani school children [34].

Supporting the statements of [34] and elaborating the reasons behind the behavioral problems of children, Asad *et al.* (2017) and Karmaliani *et al.* (2017) emphasized that peer violence in Pakistan is rooted in poverty and the socialization of children, especially at home [59, 60]. Murshid (2018) reported that one of the reasons for victimization is poor hygiene that indicates victims' disadvantaged social class to bullies [61].

Malik and Abdullah (2017) concluded on the basis of information gathered from teachers and students that violent programs on TV, news and discussions on unemployment, underemployment, and other socio-political problems were a major source of aggression among youths in Pakistan [62]. Concerning bullying, the majority of students, as well as teachers, rated verbal bullying to be a catalyst for aggression.

Interestingly, despite low economic conditions in Pakistan, internet use has significantly increased in the past two decades. In 2001, only 1.3% of the population used the internet [63], but by 2012, Pakistan was at the top 20$^{th}$ position in the world in terms of internet users [64]. One of the major reasons for the spread of the internet is the huge competition in the ISP (Internet Service Providers) and telecommunication market. The easy availability of WIFI [65] and accessibility of smartphones at ever cheaper rates have caused the number of mobile internet users to increase consistently [66].

## Status and conditions of interventions in Pakistan's context

International investigations have indicated that countries with a higher prevalence of face-to-face traditional bullying have a high level of cyberbullying as well [67]. Cyberbullying seems to co-occur with traditional bullying [24] and interventions should be pertinent for managing both types of bullying, otherwise, several studies have shown that controlling one form of bullying can lead to the perpetrator engaging in other forms of bullying [68, 69]. Many of the interventions dealing with traditional school bullying are modified for tackling cyberbullying issues on the presumptions of similarities in both types of bullying behaviour. Both constitute unjustified aggression, based on a power imbalance, and persist over time. Repetition criteria are debated among scientists as it is not as obvious in cyberbullying as it is in traditional bullying [18, 19].

The Federal Investigation Agency (FIA) of Pakistan has reported that delinquency related to the internet is constantly rising [70]. Although statistics have unveiled a higher number of cyber risks behaviours especially in youngsters, interventions designed to control cyberbullying and consequences are substandard so far in Pakistan's context [35]. Similarly, despite a high frequency and concerns about bullying and victimization as a public health issue in low- and middle-income countries in addition to the chronicity of behavioral problems there are limited policies and interventions designed, implemented, and evaluated [34]. One of the effective trials conducted by McFarlane et al. (2017) was the application of an international intervention program named "Right to Play Intervention" [43]. In this whole-school approach students were engaged with different physical activities to help build their cognitive, social, emotional, and physical skills. Right To Play's Positive Child and Youth Development program in Pakistan includes games and activities from the manual Red Ball Child Play that focuses on

4 areas of youth development, including physical, cognitive, social, and emotional domains. However, this intervention did not produce convincing results and the authors suggested several limitations and differences in the context of Pakistan in terms of climate, living conditions, attitudes towards school, etc. Contrarily, in another study by Karmaliani et al. (2020) play-based life-skills interventions delivered in public schools in Pakistan were able to elicit a significant reduction in peer violence [71].

Maryam and Ijaz (2019) also attempted to integrate some of the activities from the anti-bullying program of Operation Respect from the USA in addition to behavioral and cognitive techniques used in therapy with school children in Pakistan [72, 73]. The program was implemented over a 4-months timespan with the main focus on enhancing the pro-social skills, emotional management and problem-solving aptitude of the victims. The participants showed improvement in the skills taught to them, and an overall reduction was seen in the incidents of bullying.

Hakim and Shah (2017) investigated strategies used to control bullying in primary schools of Haripur, Pakistan. They found that the majority of the teachers adopted the strategy of providing a safe physical environment by instructing about rules before engaging students in any activity to control bullying and behavioral problems [74]. It should be noted that teachers' job satisfaction can also be achieved by creating a conducive working environment and fostering strong relationships [75]. However, detailed information and steps for the creation of a conducive environment were not specifically discussed or elaborated in the study of Hakim and Shah (2017) [74]. Similarly, involving parents and students to stand against bullying was also reported by teachers but content, methodology, and details of the intervention programs were not provided or clarified. It is concluded that while there are general rules of understanding on how to handle bullying issues, expertise in this field is still insufficient.

This review of the limited number of interventions in Pakistan has shown that there is a need for intervention of bullying and cyberbullying. Moreover, most of the interventions adapted/adopted and applied were only focusing on one aspect of training like engaging with physical activities [43, 71], creating safe physical environment [74], enhancing pro-social skills or emotion regulation of the victims [72]. Naveed et al. (2020) have further emphasized the need to comprehend the underlying patterns of behavioral difficulties in order to devise effective pragmatic anti-bullying initiatives, school-based mental health services and psychosocial counseling procedures [34]. Using a literature review, which focuses on Pakistan's particular context, the authors of the current study conducted a baseline survey to get a snapshot of what teachers believe about bullying incidents and what interventions they expect to be able to use to identify, combat, and stand up to bullying. Educators have a broader role to play than just in the classroom; they can contribute to the overall planning and implementation of schools' policies and plans [76]. Teachers are the primary agents that can influence the entire school environment by introducing measures against bullying perpetration and victimization [77]. A constant presence of teachers in classrooms throughout the academic year also allows students to seek help whenever they experience or witness bullying or victimization. When designing a teacher-led intervention, it is essential to obtain information about bullying in educational institutions from teachers and to ask their opinions about the intervention design they will use to address bullying. With this goal in mind, this study was designed to obtain the necessary information from teachers before designing and implementing a teacher-led intervention program.

## Methods

The present study used a quantitative cross-sectional survey design to determine the prevalence of different forms of bullying in educational institutions of Pakistan. Many of the referenced

researchers noted that the prevalence of bullying and its subtypes in educational institutions in Pakistan is understudied [6, 34, 35, 49, 50]. The purpose of this study was to fill this gap by expanding the knowledge about bullying and its categories in Pakistani educational institutions.

## Ethical statement

The researchers followed basic ethical principles and the APA's ethical code. Participants provided informed consent through an online forum which can be considered written consent since, after reading details about the purpose of the study, anonymity, free participation, planned use of data, and the right to end participation without negative consequences, they explicitly agreed to these study conditions by clicking the "Agree" button at the beginning of the online survey. In this way, informed consent was assured according to the ethical guidelines and federal legislation. Participants were adults and the questionnaire did not relate to their own victimization experiences thus the probability of re-victimization was low. Ethical review and approval were not required for the study on human participants in accordance with the local legislation and institutional requirements. The entire study and questionnaire were reviewed by the second author's research team consisting of educational psychologists and educationalists and two educators from a private university in the Metropolis City of Pakistan who are well acquainted with the educational system of Pakistan. The team of reviewers found no potential conflict of interest or harm to participants, nor any activities that went beyond the ethical code of conduct.

## Instruments

The study variables were assessed using a questionnaire in which teachers reported the prevalence of bullying among students as they perceived it.

**Demographics.** Participants' demographic information was used to determine their level of teaching (e.g. primary, secondary), type of institution (e.g. public, private) and place of institution (urban, rural).

In order to assess *cyberbullying* from the teachers' perspectives, an adapted version of the Berlin Cyberbullying-Cybervictimization Questionnaire (BCyQ) by Schultze-Krumbholz and Scheithauer (2011) was used [78]. Teachers were asked 17 statements on a 5-point Likert scale (1 = never to 5 = frequently) to identify their perceptions of cyberbullying incidents among students (*Example statements*: *"Bad things were told/written about student on the Internet or by mobile phone to destroy his/her friendships or reputation."*, *"Student received messages on the Internet or by mobile phone in which he/she was verbally abused or insulted."*).

To assess *social bullying* among students from the teachers' perspective researchers adapted and contextualized 10 items developed by Doğruer (2015) [79]. A five-point Likert scale was used (1 = never to 5 = frequently). (*Example statements*: *"Some student(s) prevent other students from being friends with people they don't like."*, *"Some student(s) tell lies and stories about others students to make them look bad.")*.

*Verbal bullying* among students was measured using adapted and contextualized items from various studies [79–81]. Eight items with a five-point Likert scale were used (1 = never to 5 = frequently). (*Example statements*: *"Some student(s) swears at others."*, *Some student(s) has insulted or said nasty words to others."*, *"Some student(s) threatens to physically hurt someone.")*.

Nine items were adapted and contextualized from various studies [80–82] to measure *physical bullying* among students on a five-point Likert scale (1 = never to 5 = frequently). (*Example statements*: *"Some student(s) has thrown things at another student or hit others with an object for physical abuse."*, *"Some student(s) has tripped (causing someone to stumble or fall) another student on purpose.")*.

**Table 1. Demographics.**

| Demographic Variable | | No. of Participants | Percentage |
|---|---|---|---|
| Level of Teaching | Primary (1–5 Grades) | 101 | 22.2 |
| | Secondary (6-10/ O-Levels) | 186 | 41.0 |
| | Higher Secondary (A-Levels) | 60 | 13.2 |
| | University | 107 | 23.6 |
| Institutional Location | Urban | 364 | 80.2 |
| | Sub Urban | 27 | 5.9 |
| | Rural | 63 | 13.9 |
| Institutional Setting | Public | 129 | 28.4 |
| | Semi Private/Public | 56 | 12.3 |
| | Private | 269 | 59.3 |
| | Total | 454 | 100.0 |


*Teachers' opinions about new antibullying interventions* were also measured with six items to better understand what educators expect from new interventions. Identifying the characteristics of an intervention that teachers expect to help in controlling bullying in their schools was the primary goal. Each item was based on a single question (Refer to Table 6).

The questionnaire was reviewed by the research team of the second author consisting of educational psychologists and educational scientists and two educators from a private university in the Metropolis City of Pakistan who are well acquainted with Pakistan's educational system. Several items were revised by the German and Pakistani experts in order to avoid ambiguous statements, to eliminate duplicate or compound questions, and to contextualize and adjust statements for better understanding by Pakistani teachers.

### Participants

Using Google forms, a questionnaire survey was conducted online. Over 1,000 forms were sent to educators to invite them to participate in the study. The researchers followed basic ethical principles and the APA's ethical code. Teachers were informed about the study's purpose, voluntariness of participation and were given the right to withdraw from the study. In total, 454 teachers from different parts of Pakistan responded to the questionnaire from November 2021 until January 2022. The demographics of the participants are summarized in Table 1.

Most of the responding teachers were from secondary-level education (41%), from urban settings (80.2%), and associated with private educational institutions (59.3%). Also, teachers were asked about the source of information concerning bullying incidents (see Table 4) and the most common plattform for cyberbullying (see Table 5).

## Data analysis

Data were analyzed using one-way ANOVA and Pearson correlation in SPSS 27. Exploratory and confirmatory factor analysis were computed using SPSS 27 and AMOS 22. To determine if the sample data is drawn from a normally distributed population (within a certain tolerance), a normality test is usually performed. Several statistical tests require normally distributed sample populations, such as the student's t-test and the one-way and two-way ANOVA. Normality can have serious effects in small samples, but this impact effectively diminishes when sample size reaches 30 according to Cohen et al. (2002) and 50 according to de Winter et al. (2009) [83, 84]. This means that the sampling distribution of the mean can be

assumed to be normal if each sample contains a large number of observations (in the present study $n = 454$).

## Factor analysis

Pre-analyses indicated that the sample size is satisfactory as the KMO value is higher than 0.7 (.961) [85, 86] and an exploratory factor analysis can be conducted as the Bartlett's test is significant ($\chi^2(630) = 10533.685$, $p < .001$) [87, 88]. Results of the factor analysis and factor loadings are shown in Table 2. After the factor analysis some of the items were removed to satisfy model fit criteria and reliability indexes. The final questionnaire was based on 36 questions

**Table 2. Factor loadings and reliability statistics.**

| S. No | Type of Bullying | Estimates | No. of Items | Reliability Cronbach Alpha |
|---|---|---|---|---|
| 1. | Cyber | .591 | 13 | .931 |
| 2. | Cyber | .619 | | |
| 3. | Cyber | .719 | | |
| 4. | Cyber | .695 | | |
| 5. | Cyber | .728 | | |
| 6. | Cyber | .715 | | |
| 7. | Cyber | .768 | | |
| 8. | Cyber | .741 | | |
| 9. | Cyber | .713 | | |
| 10. | Cyber | .726 | | |
| 11. | Cyber | .778 | | |
| 12. | Cyber | .768 | | |
| 13. | Cyber | .731 | | |
| 1. | Social | .667 | 9 | .898 |
| 2 | Social | .652 | | |
| 3 | Social | .706 | | |
| 4 | Social | .707 | | |
| 5 | Social | .677 | | |
| 6 | Social | .764 | | |
| 7 | Social | .739 | | |
| 8 | Social | .736 | | |
| 9 | Social | .726 | | |
| 1 | Verbal | .699 | 7 | .900 |
| 2 | Verbal | .750 | | |
| 3 | Verbal | .759 | | |
| 4 | Verbal | .768 | | |
| 5 | Verbal | .746 | | |
| 6 | Verbal | .685 | | |
| 7 | Verbal | .785 | | |
| 1 | Physical | .738 | 7 | .919 |
| 2 | Physical | .791 | | |
| 3 | Physical | .801 | | |
| 4 | Physical | .757 | | |
| 5 | Physical | .796 | | |
| 6 | Physical | .785 | | |
| 7 | Physical | .787 | | |


**Table 3. Prevalence of different bullying types as perceived by teachers.**

| | Cyberbullying | Verbal Bullying | Physical Bullying | Social Bullying |
|---|---|---|---|---|
| N | 454 | 454 | 454 | 454 |
| Mean | 2.6415 | 3.1718 | 2.7942 | 3.3529 |
| Standard Deviation | .87115 | .85354 | .91081 | .81194 |
| Range | 4.00 | 4.00 | 4.00 | 4.00 |


with 4 main factors (Cyberbullying- 13 items, Social Bullying- 9 items, Verbal Bullying-7 items, Physical Bullying-7 items) (Refer to Table 2).

For every construct, Cronbach's is higher than 0.7, indicating that the subscales are reliable [89] (Refer to Table 2). Construct validity is established through Average Variance Extracted (AVE) which was 0.60 and can be considered as good [90]. Further indicators of model fit also show that the instrument meets the model fit criteria ($\chi^2$/df = 2.650, CFI = .904, RMR = .066, RMSEA = .054, AGFI = .815, IFI = .905, PCFI = .810, PNFI = .716).

# Results

## Prevalence of bullying in educational institutions

The results regarding the frequency of different types of bullying incidents are shown in Table 3. The purpose of compiling this information is to determine how many teachers have witnessed bullying incidents among students and shared their experiences. From the Table 4, it appears that teachers have witnessed more social and verbal bullying incidents (mean values are higher than 3) than physical and cyberbullying.

The first common source of information is "sharing between a teacher and a colleague" (see Table 4). In addition, the majority of reported cases involved teachers themselves witnessing incidents and documenting them in surveys. The third common source of information was that the victim himself/herself reported it to the teacher. This shows that peer bystanders are least likely to speak out about the incident as compared to victimized children.

Additionally, which cyber platform is most commonly used for cyberbullying was also collected and is shown in Table 5. Participants were asked to respond to the question: In terms of cyberbullying incidents which is the most common networking site students are using? The information gathered indicates that Facebook is still the most commonly used network for cyberbullying, followed by Instagram and TikTok.

**Table 4. Source from which teachers get information about bullying incidents.**

| | | Frequency | Percent |
|---|---|---|---|
| | Never Witnessed | 6 | 1.3 |
| | Witnessed by Teacher | 113 | 24.9 |
| | Reported by Victim | 80 | 17.6 |
| | Reported by Peer Bystander | 46 | 10.1 |
| | Heard from Colleagues | 121 | 26.7 |
| | Anonymous information | 52 | 11.5 |
| | Reported by Parent | 32 | 7.0 |
| | Various Means | 4 | .9 |
| | Total | 454 | 100.0 |


Table 5. Cyberbullying forums.

| Responses | | Frequency | Percent |
|---|---|---|---|
| Valid | No Idea | 8 | 1.8 |
| | FaceBook | 251 | 55.3 |
| | TikTok | 55 | 12.1 |
| | Whatsapp | 10 | 2.2 |
| | Instagram | 69 | 15.2 |
| | Snapchat | 14 | 3.1 |
| | YouTube | 26 | 5.7 |
| | Twitter | 11 | 2.4 |
| | Zoom | 2 | .4 |
| | All | 6 | 1.3 |
| | Gaming Apps | 2 | .4 |
| | Total | 454 | 100.0 |


## Association of bullying with level of teaching and educational setting

To understand the association between the level of teaching and different forms of bullying, the Pearson correlation test was run (Refer to Table 6). There is a significant decrease in the cases of physical bullying as the teaching level increases, which indicates that the physical form of bullying is more prevalent in primary schools than at higher educational institutions. In addition, cyberbullying is positively correlated with traditional forms of bullying, suggesting that institutions where cyberbullying is prevalent also experience higher levels of physical, verbal, and social bullying.

To examine the difference between educational institutional settings (independent variable) regarding different forms of bullying (dependent variable), one-way ANOVA tests were conducted (Refer to Tables 7 and 8). Using one-way ANOVA, bullying incidents among students were compared by locality of the educational institution (urban, sub-urban, and rural). According to the analysis, there was a significant difference for social bullying ($F(2,451) = 7.419$, $p = 0.001$). However, no significant differences were found for physical, verbal, or cyberbullying. According to a post-hoc analysis using the Tukey method, there was a significant difference in the prevalence of social bullying in rural ($M = 3.004$, $SD = 0.74615$) and urban institutions ($M = 3.42$, $SD = 0.81050$). From mean differences, we found that urban teachers reported more cases of social bullying than their rural counterparts (see Table 8). In this study, teachers from urban schools were more likely to respond ($N = 364$) compared to teachers from rural schools ($N = 63$), leading to uneven group sizes. The discussion section outlines some other possible explanations for these differences. Similarly, one-way ANOVA was also

Table 6. Correlations.

| | Level of Teaching | Cyberbullying | Verbal Bullying | Physical Bullying | Social Bullying |
|---|---|---|---|---|---|
| Level of Teaching | 1 | .052 | -.016 | -.113* | .001 |
| Cyberbullying | | 1 | .633** | .572** | .628** |
| Verbal Bullying | | | 1 | .677** | .795** |
| Physical Bullying | | | | 1 | .565** |
| Social Bullying | | | | | 1 |

*. Correlation is significant at the 0.05 level (2-tailed).

**. Correlation is significant at the 0.01 level (2-tailed).


**Table 7. ANOVA.** Dependent Variable: Form of Bullying.

| | | Sum of Squares | Df | Mean Square | F | Sig. |
|---|---|---|---|---|---|---|
| Cyberbullying | Between Groups | 1.013 | 2 | .507 | .667 | .514 |
| | Within Groups | 342.772 | 451 | .760 | | |
| | Total | 343.786 | 453 | | | |
| Verbal Bullying | Between Groups | 3.161 | 2 | 1.580 | 2.181 | .114 |
| | Within Groups | 326.867 | 451 | .725 | | |
| | Total | 330.028 | 453 | | | |
| Physical Bullying | Between Groups | .230 | 2 | .115 | .138 | .871 |
| | Within Groups | 375.564 | 451 | .833 | | |
| | Total | 375.794 | 453 | | | |
| Social Bullying | Between Groups | 9.512 | 2 | 4.756 | 7.419 | .001 |
| | Within Groups | 289.129 | 451 | .641 | | |
| | Total | 298.641 | 453 | | | |

Independent Variable: Institutional Location (Urban, Rural, Sub Urban)


conducted to examine differences between bullying incidents (dependent variable) in public, private and semi-private institutions (independent variables). No significant difference was observed.

### Teachers' opinions about new intervention

Teachers were also asked about various aspects of new interventions to control traditional bullying and cyberbullying. Table 9 provides an overview of the items and participants' responses. Finally, recommendations and suggestions offer further insight into teachers' opinions.

## Discussion

In this research paper, the authors collected information about bullying incidents among students observed and reported by teachers. It becomes evident that teachers have noticed more social and verbal bullying incidents than physical bullying and cyberbullying. The opposite has been found in other studies when data was collected from students in Pakistan, which showed that traditional and cyberbullying was a frequent practice in Pakistan's educational institutions [34, 35]. Investigations have shown that bullying and cyberbullying occur regularly in Pakistan's educational institutions, which negatively affect the mental, emotional, and physical

**Table 8. Multiple comparisons.**

**Tukey HSD**

| Dependent Variable | (I) Institutional Location | (J) Institutional Location | Mean Difference (I-J) | Std. Error | Sig. | 95% Confidence Interval | |
|---|---|---|---|---|---|---|---|
| | | | | | | Lower Bound | Upper Bound |
| Social Bullying | **Urban** | Sub Urban | .15223 | .15970 | .607 | -.2233 | .5278 |
| | | **Rural** | **.41619*** | **.10926** | **.000** | .1593 | .6731 |
| | Sub Urban | Urban | -.15223 | .15970 | .607 | -.5278 | .2233 |
| | | Rural | .26396 | .18417 | .325 | -.1691 | .6970 |
| | **Rural** | **Urban** | **-.41619*** | **.10926** | **.000** | -.6731 | -.1593 |
| | | Sub Urban | -.26396 | .18417 | .325 | -.6970 | .1691 |

*. The mean difference is significant at the 0.05 level.


**Table 9. Teachers' opinions about new intervention design.**

| S. No | Question | Options | Participants | % |
|---|---|---|---|---|
| 1. | In order of your preference tick the most suitable mode of training intervention? | Face to Face only | 250 | 55.1 |
| | | Online only | 38 | 8.4 |
| | | Blended | 99 | 21.8 |
| | | No Preference (All options work) | 67 | 14.8 |
| 2. | Do you think that installing apps for bullying prevention can help you to learn better and utilize your free time more effectively? | Strongly Disagree | 34 | 7.5 |
| | | Disagree | 39 | 8.6 |
| | | Neutral | 134 | 29.5 |
| | | Agree | 141 | 31.1 |
| | | Strongly Agree | 106 | 23.3 |
| 3. | Bullying and Cyberbullying are associated with some serious social issues such as sexual harassment, spreading of unethical content, grooming, etc. Intervention would also be helpful for you to understand and stand against this immoral, unethical, and antireligious content. Would it be acceptable to you to have a trainer of a different gender than you and be able to openly discuss these issues? | Strongly Disagree | 14 | 3.1 |
| | | Disagree | 30 | 6.6 |
| | | Neutral | 127 | 28 |
| | | Agree | 119 | 26.2 |
| | | Strongly Agree | 164 | 36.1 |
| 4. | After training would it be feasible for you to train your students in segregated settings (boys and girls separately)? | Strongly Disagree | 17 | 3.7 |
| | | Disagree | 31 | 6.8 |
| | | Neutral | 92 | 20.3 |
| | | Agree | 135 | 29.7 |
| | | Strongly Agree | 179 | 39.4 |
| 5. | In order to control unacceptable behavior, support from holy books, prophets' lifestyles examples, and stories from religious and cultural perspectives should also be part of the intervention | Strongly Disagree | 12 | 2.6 |
| | | Disagree | 25 | 5.5 |
| | | Neutral | 78 | 17.2 |
| | | Agree | 89 | 19.6 |
| | | Strongly Agree | 250 | 55.1 |
| 6. | Discussion with other ethnicities, nationalities, and cultural exchange on social networking platforms (skype, zoom, or google meet) would be effective in reducing bullying/cyberbullying. | Strongly Disagree | 22 | 4.8 |
| | | Disagree | 32 | 7 |
| | | Neutral | 99 | 21.8 |
| | | Agree | 132 | 29.1 |
| | | Strongly Agree | 169 | 37.2 |


health of Pakistan's children and young adults [35, 37, 48]. Khawaja et al. (2015) found physical and verbal violence as a prevalent occurrence in major cities and provincial capitals' institutions of Pakistan [44]. However, the present data collected from teachers do not support the claim that bullying is prevalent in schools in the form of cyber or physical afflictions. Teachers' lack of interaction with students on social networking platforms where cyberbullying happens can be a contributing factor for deviating reports of cyberbullying. In addition, teachers usually aren't members of groups on social networking sites where students engage in cyberbullying and, since students maintain anonymity, teachers can't report or respond to students engaging in cyberbullying. Aside from that, the involvement in continuous online teaching platforms has reduced the time available for teachers to interact with students for socializing purposes or networking. Studies have shown that during the pandemic, online teaching-related technology stress suppresses the urge among teachers to interact online for purposes other than those related to education [91]. As a result of these contradictory statements, it can be concluded that bullying and cyberbullying are prevalent in educational institutions, but teachers are infrequently exposed to those incidents or are unfamiliar with them, and need training to identify

victimization and perpetrator behavior to intervene effectively and control unacceptable behavior.

Research has revealed that physical bullying tends to decrease as one goes up the educational ladder. This means physical bullying is more prevalent in primary schools than in higher education institutions. Zych *et al*. (2020) found that physical bullying decreases throughout adolescence, which is consistent with these results [92]. Contrary to Rafi et al. (2019), no significant difference was found between cyberbullying incidents and age in the present study [38]. Our results further demonstrate that cyberbullying is positively correlated with verbal, physical and social forms of bullying in institutions with a higher prevalence of cyberbullying, these forms of bullying are occurring more commonly. These results are in line with the explanation given by Kowalski et al. (2014) who stated that cyberbullying usually occurs in conjunction with traditional bullying, and institutions where cyberbullying is common also have a higher incidence of traditional bullying incidents [24].

Bjereld (2018) argued that victims often fear being seen as victims by others and thus attempt to conceal their victimization [93]. Victims are usually unenthusiastic to share their suffering due to multiple reasons such as distrust of adults and concern about being blamed [93]. On the contrary, the present teacher data revealed that bystanders are the least likely to report an incident. Bystanders are often the first to witness an incident and report it to a teacher, which is why in most interventions they are a central target of training [93]. However, the number of bystanders reporting an incident to a teacher or intervening directly appears to be lowest in Pakistan. Gordon (2019) reported numerous reasons why bystanders do not intervene or communicate incidents to adults [94]: fear of being victimized for reporting or intervening, a lack of knowledge of what to do in such a situation, mistrust of adults, having been taught to stay away from this kind of situations, and moral disengagement beliefs. There is still a need to investigate the reasons for the silence of bystanders in Pakistan. Additionally, the bystanders' role in the bullying chain should be emphasized in upcoming intervention designs for Pakistan's educational institutions.

Many of the social networking sites have been used as a venue for cyberbullying activities such as Instagram [95], twitter [96], Facebook [97], TikTok [98], WhatsApp [99], Snapchat [100], YouTube [101] Zoom [102], and online games [103]. However, our data from teachers in Pakistan showed that Facebook continues to be the most frequently used social networking platform where cyberbullying incidents are observed. There is a possibility that other platforms that are gaining more popularity among young people are being misused for cyberbullying. Teachers, however, still preferably use Facebook and therefore haven't noticed such incidents on other social networks. In order to find out the popularity of emerging social networks among youths, data should also be collected from them to reveal the true picture of social networking. Moreover, despite efforts being made by social networking sites to control bullying, there is still a considerable prevalence of bullying that indicates that users need training to protect themselves and others from cyber harms of online networking.

As Tayyaba (2012) demonstrates, there are differences between rural and urban educational institutions that are related to differences in student performance [104]. Different parenting styles, social characteristics and school climate may be responsible for this. As a result, the opinions and experiences of teachers may also differ regarding bullying issues. However, this area of research is still understudied and requires more comprehensive and detailed investigations. Moreover, a baseline assessment should always be conducted so that differences between rural and urban settings can be taken into account and bullying interventions can be tailored to the needs of individual schools.

## Recommendations and suggestions

The previous sections already discussed that there are limited interventions designed to control traditional bullying in schools and hardly any intervention for controlling cyberbullying successfully implemented in Pakistan. It is now a major concern whether adopting Western school-based bullying control interventions would be promising in South Asia. Moreover, it is a question whether overburdened teachers will be able to participate and implement the program successfully. Based on the review of the literature and the baseline survey from teachers there are a few suggestions made by the authors for addressing this problem in Pakistan's educational institutions.

1. McFarlane et al. (2017) reported that Pakistan is a particularly challenging country for evaluating international interventions because of multiple variations in terms of climate and school cultures [43]. Given the specific societal, political, economic, and climatic challenges that teachers and students encounter in Pakistan [43], school-based interventions cannot produce successful results unless combined with some other measures. It is possible to introduce web-based interventions through teachers' professional development. Web-based interventions are also considered a cost-effective strategy, which can maintain anonymity/privacy and can address a large number of people [105]. These reports were cross-examined by asking about the teachers' preferred mode of training (blended, online or face-to-face training). A surprising 55.1% of the participants wanted to meet instructors in person and preferred face-to-face training sessions. Due to the difficulties Pakistani teachers faced during the transition to online learning and teaching, this phase has left a negative impression on teachers about online instruction. The failure of online learning to produce effective results has been attributed to multiple reasons [106–108]. The challenges of digital transformation of the educational system include poor internet signals/strength, high internet connectivity costs, electricity load shedding, lack of training and readiness of remote learners, difficulty in group activities, unreliable assessment methods and results, and insufficient interaction among participants and instructors [106, 108]. Some of the students reported health concerns from continuous online learning such as eye sight and ear pain issues [50]. Beside the lack of immediate feedback, the practical component of learning is also cited as a major problem [107]. In light of this observation, the authors suggest that teachers receive on-site training for anti-bullying intervention.

2. Cyberbullying is usually accompanied by traditional bullying [24] and it is recommended that interventions designed should be pertinent to managing both types of bullying, otherwise there are studies where controlling one form of bullying results in involvement of the perpetrators in another type of bullying [68, 69]. The statistical results of the current study have also revealed that students at institutions where cyberbullying occurs more frequently are also at greater risk of traditional bullying such as physical, verbal, and social forms. From these interpretations, it is concluded that new intervention programs should not only address traditional bullying but should also target cyberbullying. In addition, literature has shown both cyberbullying and traditional bullying are prevalent in Pakistan, but the teachers' data explained that the prevalence of cyberbullying or physical assault is less common than other two forms of bullying. These contradictory statements highlight the prevalence of bullying and cyberbullying in educational institutions, but educators are rarely exposed to these incidents or are unfamiliar with them. Therefore, educators need training in detecting victimization and perpetrator behaviors. A new intervention should include components that will assist teachers in identifying victimization and perpetration, allowing them to intervene immediately and control the situation.

3. The use of mobile apps and Virtual realities (VR) as a combating strategy is also suggested by many researchers. Apps such as Shazam or Unmute Daniel and iZ Hero are some of the technology-based interventions designed to create awareness and prevent bullying [109, 110]. The introduction of such programs can be effective in controlling behavioural problems with limited financial resources. Despite technological challenges, 54% of teachers in our study agreed that the use of apps, in combination with face-to-face sessions, can result in more effective training. Out of 454 respondents, 134 supported a neutral position. As a result, researchers suggest creating an app that supports intervention training, where participants can continue to learn at their own pace without being restricted by geography.

4. Maddison (2013) has clearly indicated that the social structure of South Asia is more complex than elsewhere [111]. Moreover, talking about cyber-associated risky behaviours like stalking and sexting is challenging due to cultural, moral, and religious beliefs in the Pakistani community [112]. Similarly, many families are against co-education and prefer to send their children to single-sex institutions [113]. Keeping the religious and cultural aspects in mind, teachers were also surveyed regarding their acceptance of segregated settings (men and women separated) for training sessions. Unexpectedly, 283 out of 464 participants are comfortable if they are professionally trained in a group setting with tutors of a different gender as them. It shows that despite religious and cultural beliefs, teachers are ready to study with male or female counterparts. On the other hand, when their perceptions on transferring knowledge to students after training was assessed, almost two thirds of the participants thought segregated classes would provide opportunities for both girls and boys to speak more openly about issues and concerns (if any) with instructors and fellow students. The implementation of separate programs for men and women is also recommended to ensure open dialogue and discussion without hurting any community's religious or traditional beliefs.

5. Religious education is widely used in Pakistan for character building and moral engagement. Previous studies have established that religious education contributes to moral development, like Perrin (2000), who found a positive association between honesty and religiosity [114]. Nucci (1989) stated, on the contrary, that children should be taught universal values devoid of ethnic and geographical differences in order to live a meaningful life [115]. He argued that when children are controlled by religious doctrines, they do not adopt these values, rather they rebel and refuse. In order to make an informed recommendation, the present study also asked teachers about their opinion on taking support from holy books, prophets' behavioral examples, and stories from religious and cultural perspectives to improve children's behavior. According to the descriptive results, three-fourths of participants agreed that religious support was necessary for the new intervention to be effective.

6. Khawaja *et al.* (2015) suggested that recreational and cultural enrichment programs can be beneficial to pupils, because they are exposed beyond the boundaries of their own community which may create tolerance and the motivation needed to improve their circumstances and behaviors [44]. Our descriptive findings revealed that about 66% of the teachers believed that forums in which students interact with other cultures and share their experiences would help them to develop tolerance towards others, acceptance of different opinions, and understanding of other cultures. As such, it is recommended that any future interventions emphasize some of the components needed to develop better intercultural perspectives and focus on cultural intelligence.

7. Peer training involves students either of the same age or of different ages who learn from each other in a structured manner. This type of training allows students to put their

knowledge and skills to use, and creates a platform for both the trainer and trainee to boost their self-confidence. The tutor gains confidence in their ability to assist someone, while the trainee receives encouragement from their peers, strengthening their sense of self-competence [116]. It is recommended that along with teachers' professional development, components for peer training should also be introduced to control bullying incidents in educational settings.

## Practical implications

This research study served as a baseline assessment conducted prior to the development of a teacher-led anti-bullying training program and has several practical implications. An analysis of the survey provided a comprehensive understanding of bullying in Pakistani schools and educational institutions. This information has been used to determine the extent of the problem and the need for an anti-bullying training program. The study also revealed what types of bullying are most common and what teachers expect from new interventions. This information can then be used by future researchers to tailor anti-bullying training programs to the specific needs of students.

## Conclusions

This study serves as the basis for an anti-bullying intervention developed for a teacher training course. It is novel in that it does not rely on student self-reports of bullying incidences and frequencies as most of the previous studies in Pakistan have done and it includes teacher perceptions and experiences. The results show some clear differences to the research conducted previously in Pakistan.

A review of literature has already demonstrated that bullying and cyberbullying are undeniably prevalent in Pakistan's educational system and society and there is a dire need to develop and integrate bullying interventions. The current study also confirmed that verbal and social forms of bullying are widespread from the teachers' point of view. This study has provided a new perspective and general recommendations for planning and implementing new, socially contextualized anti-bullying interventions for teachers in Pakistan. According to the baseline survey, the intervention should not only explain how to handle and control bullying, but it should also provide training for teachers to help them identify victimization and perpetrator behavior, both in traditional schools and online. Moreover, it is equally imperative that intervention should be focused on both traditional and cyber forms of bullying, as both of these are prevalent in educational institutions. In Pakistan, bystanders are least likely to intervene or report bullying to teachers, though they are considered the strongest link in the bullying chain. It is also helpful to train peers or bystanders to intervene as soon as a bullying incident is observed.

The study findings have assisted the authors in developing a low-cost antibullying program led by teachers. Teachers will receive professional development in addressing bullying in institutions to implement the intervention design based on the outcomes of this study. An important concern is whether a teacher-led intervention can empower students to reduce their negative behavior in the context of poverty and social norms supportive of violence. To ensure the effectiveness of the new intervention design, further research is recommended for the future.

## Limitations and directions for future research

Despite its strengths, the study also has certain limitations. Increasing the sample size and including additional educational stakeholders such as administrators, counselors, and

educational institution staff could provide a better insight into the issues. In addition, translated versions of the instruments are recommended, especially for teachers from rural areas where English is not commonly used for academic or communication purposes. A major limitation is the cross-sectional design of the study, which generally does not accurately capture actual measurement invariance scores over time. It is therefore suggested that similar studies be repeated with a longitudinal design, changing the sample size and including data from more institutions in different regions of Pakistan.

## Author Contributions

**Formal analysis:** Sohni Siddiqui.

**Investigation:** Sohni Siddiqui.

**Methodology:** Sohni Siddiqui.

**Project administration:** Anja Schultze-Krumbholz.

**Supervision:** Anja Schultze-Krumbholz.

**Writing – original draft:** Sohni Siddiqui.

**Writing – review & editing:** Anja Schultze-Krumbholz.

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
