## [Decision Letter · Decision Letter 0]

17 Jan 2023

PONE-D-22-34325Bullying prevalence in Pakistan’s educational institutes and recommendations for tailored interventions: Preclusion to the framework for a teacher-led antibullying interventionPLOS ONE

Dear Dr. Siddiqui,

Thank you for submitting your manuscript to PLOS ONE. After careful consideration, we feel that it has merit but does not fully meet PLOS ONE’s publication criteria as it currently stands. Therefore, we invite you to submit a revised version of the manuscript that addresses the points raised during the review process.

You are required to complete the revisions. 

We look forward to receiving your revised manuscript.

Kind regards,

Faisal Shafique Butt

Academic Editor

PLOS ONE

Journal Requirements:

Additional Editor Comments:

You are required to complete the minor revisions.

Reviewers' comments:

Reviewer's Responses to Questions

**Comments to the Author**

1. Is the manuscript technically sound, and do the data support the conclusions?

Reviewer #1: Yes

Reviewer #2: Partly

2. Has the statistical analysis been performed appropriately and rigorously? 

Reviewer #1: Yes

Reviewer #2: Yes

3. Have the authors made all data underlying the findings in their manuscript fully available?

Reviewer #1: Yes

Reviewer #2: Yes

4. Is the manuscript presented in an intelligible fashion and written in standard English?

Reviewer #1: Yes

Reviewer #2: No

5. Review Comments to the Author

Reviewer #1: The manuscript is a good attempt but a few changes may improve the overall quality of the research study.

1. Title of the research study is too lengthy, therefore, it needs to be shortened.

2. The citations should be properly done. For instance, [17] enumerated that the level of cyberbullying has substantially increased in educational 95 institutions in Pakistan. Before [17] the name of the author should be properly written in the correct style.

3. To enrich the literature Review, cite the following relevant studies

https://papers.ssrn.com/sol3/papers.cfm?abstract_id=2599095

https://papers.ssrn.com/sol3/papers.cfm?abstract_id=2600403

https://papers.ssrn.com/sol3/papers.cfm?abstract_id=2602997

https://papers.ssrn.com/sol3/papers.cfm?abstract_id=2602965

https://papers.ssrn.com/sol3/papers.cfm?abstract_id=2602568

3. Recommendations and Limitations should be written separately under the conclusion.

Reviewer #2: The author must remove the pin pointing flaws in the paper in order to improve its quality. These flaws are minor in nature but are very important to improve the quality of paper.

6. PLOS authors have the option to publish the peer review history of their article (what does this mean?). If published, this will include your full peer review and any attached files.

Reviewer #1: **Yes: **Dr. Najabat Ali

Reviewer #2: **Yes: **Dr. Abdul Ghafoor Awan

---

## [Author Response · Author response to Decision Letter 0]

7 Mar 2023

Comments and suggestions from reviewers have been addressed. In the file Response to reviewers comments, you can find detailed explanations of each comment and response.

---

## [Editor Report · Decision Letter 1]

13 Mar 2023

PONE-D-22-34325R1Bullying prevalence in Pakistan’s educational institutes: Preclusion to the framework for a teacher-led antibullying interventionPLOS ONE

Dear Dr. Siddiqui,

Thank you for submitting your manuscript to PLOS ONE. After careful consideration, we feel that it has merit but does not fully meet PLOS ONE’s publication criteria as it currently stands. Therefore, we invite you to submit a revised version of the manuscript that addresses the points raised during the review process.

Please fulfill reviewers comments. 

Please submit your revised manuscript by 30th March 2023. If you will need more time than this to complete your revisions, please reply to this message or contact the journal office at plosone@plos.org. Please include the following items when submitting your revised manuscript:A rebuttal letter that responds to each point raised by the academic editor and reviewer(s). You should upload this letter as a separate file labeled 'Response to Reviewers'.A marked-up copy of your manuscript that highlights changes made to the original version. You should upload this as a separate file labeled 'Revised Manuscript with Track Changes'.An unmarked version of your revised paper without tracked changes. You should upload this as a separate file labeled 'Manuscript'.If applicable, we recommend that you deposit your laboratory protocols in protocols.io to enhance the reproducibility of your results. Protocols.io assigns your protocol its own identifier (DOI) so that it can be cited independently in the future. For instructions see: https://journals.plos.org/plosone/s/submission-guidelines#loc-laboratory-protocols. Additionally, PLOS ONE offers an option for publishing peer-reviewed Lab Protocol articles, which describe protocols hosted on protocols.io. Read more information on sharing protocols at https://plos.org/protocols?utm_medium=editorial-email&utm_source=authorletters&utm_campaign=protocols.

We look forward to receiving your revised manuscript.

Kind regards,

Faisal Shafique Butt

Academic Editor

PLOS ONE

Journal Requirements:

Additional Editor Comments (if provided):

You are required to fulfill reviewers comments

---

## [Author Response · Author response to Decision Letter 1]

14 Mar 2023

This is not a lab study or registered report, so pilot study data requirement is not fulfilled. There are no other large files, figures or appendices used in the manuscript. 

The reviewer’s comments have already been addressed in my last submission but as per Journal’s requirements and editor’s comments references have been checked once again.

---

## [Editor Report · Decision Letter 2]

27 Mar 2023

PONE-D-22-34325R2Bullying prevalence in Pakistan’s educational institutes: Preclusion to the framework for a teacher-led antibullying interventionPLOS ONE

Dear Dr. Siddiqui,

Thank you for submitting your manuscript to PLOS ONE. After careful consideration, we feel that it has merit but does not fully meet PLOS ONE’s publication criteria as it currently stands. Therefore, we invite you to submit a revised version of the manuscript that addresses the points raised during the review process.

ACADEMIC EDITOR:

Please do the suggested changes 

We look forward to receiving your revised manuscript.

Kind regards,

Faisal Shafique Butt

Academic Editor

PLOS ONE
---

## [Author Response · Author response to Decision Letter 2]

8 Apr 2023

ACADEMIC EDITOR:

Please do the suggested changes 

Response:

The manuscript is not a lab study or a registered report protocol. The article is based on a survey conducted in Pakistan from teachers about their opinions and experiences of bullying in education. In this study validated and reliable questionnaires were used so piloting was not carried out before data collection and results are reported after full data collection completed.

---

## [Editor Report · Decision Letter 3]

11 Apr 2023

Bullying prevalence in Pakistan’s educational institutes: Preclusion to the framework for a teacher-led antibullying intervention

PONE-D-22-34325R3

Dear Dr. Siddiqui,

We’re pleased to inform you that your manuscript has been judged scientifically suitable for publication and will be formally accepted for publication once it meets all outstanding technical requirements.

Kind regards,

Faisal Shafique Butt

Academic Editor

PLOS ONE
---

## [Editor Report · Acceptance letter]

20 Apr 2023

PONE-D-22-34325R3 

Bullying prevalence in Pakistan’s educational institutes: Preclusion to the framework for a teacher-led antibullying intervention 

Dear Dr. Siddiqui:

I'm pleased to inform you that your manuscript has been deemed suitable for publication in PLOS ONE. Congratulations! Your manuscript is now with our production department. 

Kind regards, 

on behalf of

Dr. Faisal Shafique Butt 

Academic Editor

PLOS ONE